# Noncommunicable disease risk behaviors and protective factors among children in Samoa: Retrospective trend analysis of global school-based health surveys in 2011 and 2017

Courtney C. Choy[1]*, Siufaga Simi[2], Christina Soti-Ulberg[2], Take Naseri[2,3], Yasmmyn D. Salinas[1], Nicola L. Hawley[1]

1 Department of Chronic Disease Epidemiology, Yale School of Public Health, New Haven, Connecticut, United States of America, 2 Ministry of Health, Motootua, Apia, Samoa, 3 Department of Epidemiology, School of Public Health, Brown University, Providence, Rhode Island, United States of America

* courtney.choy@yale.edu

**Data Availability Statement:** All de-identified data, questionnaires, codebooks, and other study

## Abstract

Pacific Island countries experience a high prevalence of noncommunicable diseases (NCDs), which may be prevented by reducing risk behaviors and strengthening protective factors in childhood and adolescence. To better inform preventative interventions, our objective was to use publicly available data from the Global School-based Student Health Survey (GSHS), to provide cross-sectional and trend estimates for the prevalence of NCD risk and protective factors among school-aged children in 2011 and 2017 in Samoa. Two waves of cross-sectional data included 4,373 children (51.98% female), with a median age of 15 years, who were mainly in school years 9–10 in Samoa. Retrospective analyses were adjusted for the GSHS multistage stratified cluster sample design. Weighted prevalences of overweight/obesity, dietary behaviors, physical activity, and sedentary behavior, oral and hand hygiene, emotional and mental health, and community protective factors were reported by study year. Logistic regressions were fitted to assess differences in the prevalence of risk and protective factors, adjusted for age group, sex, and school year. In 2011 and 2017, the prevalence of overweight/obesity remained consistently high in females (59.12% and 64.29%, p = 0.428) and increased from 44.21% to 53.65% in males (p = 0.039). Time spent sitting for long periods, smoking cigarettes, using other tobacco products, and drinking alcohol were lower in 2017 compared to 2011 (all p<0.05). Many children reported experiencing bullying (33.27% for females and 59.30% for males in 2017), while physical fighting was common among males (73.72% in 2011 and 57.28% in 2017). The high prevalence of obesity and related NCD risk factors require urgent public health action in Samoa. Alongside the continued reduction of tobacco and alcohol use, emotional and mental wellness should be prioritized in interventions and programs to promote healthy behaviors and lifestyle changes starting in childhood.

documents are publicly available and may be downloaded from the World Health Organization noncommunicable disease microdata repository (https://extranet.who.int/ncdsmicrodata/index.php/catalog/205 and https://extranet.who.int/ncdsmicrodata/index.php/catalog/773) and the United States Centers for Disease Control and Prevention websites (https://www.cdc.gov/GSHS).

**Funding:** The secondary data analyses were supported by the Fogarty Global Health Equity Scholars Program (FIC D43TW010540). CCC is now supported by the National Heart, Lung, And Blood Institute of the United States National Institutes of Health under Award Number K99HL166781. The funders had no role in the study design, data collection and analysis, decision to publish, or preparation of the manuscript. Its contents are solely the responsibility of the authors and do not necessarily represent the official views of United States National Institutes of Health.

**Competing interests:** The authors have declared that no competing interests exist.

## Introduction

More than 70% of premature deaths in Pacific Island countries are attributed to noncommunicable diseases (NCDs) and linked to modifiable risk factors that emerge in childhood and adolescence, providing a window of opportunity for prevention and intervention [1, 2]. Starting in 2011, the United Nations General Assembly adopted a political declaration that committed member states to NCD prevention and control [3]. Subsequently, countries agreed to adopt nine global targets to reduce premature deaths from four major NCDs (cardiovascular diseases, chronic respiratory diseases, cancers, and diabetes) by 25%, compared to their 2010 levels by 2025 [4]. This Global Action Plan for NCD prevention and control has been updated to extend to 2030 and continues to focus on specific risk factors including overweight/obesity, healthy diets with lower sodium, physical inactivity, tobacco use, and harmful alcohol use [4]. Factors that were not previously prioritized in these global frameworks but are now known to impact both short-term and long-term NCD risk include emotional and mental health [5], oral health [6], and hand hygiene [7]. Loneliness, anxiety, and suicide ideation may contribute to greater risk among school-aged children and adolescents, while brushing teeth daily, washing hands, peer support, parental supervision, and connectedness can help to mitigate risk [2, 8]. Monitoring and evaluating trends in these health risks and protective factors is critical to better prioritize preventative efforts.

An ongoing collaborative surveillance project led by the World Health Organization (WHO) and the United States Centers for Disease Control and Prevention (CDC), the Global School-Based Student Health Survey (GSHS) is designed to help countries collect data for use in the monitoring and evaluation of youth health priorities [9, 10]. The survey has been administered repeatedly over time to better understand trends in behavioral risk and protective factors related to the leading causes of morbidity and mortality among children and adults worldwide [10]. Previous studies have pooled GSHS data from several countries to compare the prevalence of risk factors, including overweight/obesity, low fruit and vegetable intake, low physical activity, tobacco use, and harmful alcohol use [11, 12]. Based on GSHS data from 2007 to 2016, the prevalence of children and adolescents aged 11–17 years old reporting three or more NCD risk factors was higher in the Western Pacific region (44.5%, 95% CI: 42.1–47.1) compared to the global estimates (34.9%, 95% CI: 33.2–36.7) [11]. Between 2010 and 2011, in Pacific Island countries the proportion of 12-15-year-olds reporting three or more NCD risk factors was among the highest in the world, ranging from 11.6% in Fiji to 29.9% in Kiribati [12]. The burden of obesity and other preclinical markers of NCD risk are high and rising in the Pacific Islands [13], yet previously collected GSHS surveillance data has been underutilized. Previous research has documented the prevalence of multiple risk factors of NCDs and their cross-sectional associations [14, 15], but, to our knowledge, no studies have been conducted to assess trends in related health behaviors and protective factors in any Pacific Island setting.

Samoa is a middle-income Polynesian country, where obesity prevalence among children is far higher than the global average [13, 16] and the total cost of health care is predicted to nearly double per person between 2015 and 2040 [17]. Increasing access to imported, micronutrient-poor foods combined with increases in smoking, alcohol drinking, and physical inactivity are recognized as contributors to high levels of NCDs [18]. Several national strategies, policies, and programs have been put forth by the Samoan government and the Ministry of Health to address the pervasive NCD burden [19]. The 2006 Samoa Mental Health Policy described mental, physical, social, and spiritual health as 'indivisible' [20] and this has supported the implementation of multi-sectoral health initiatives and holistic approaches to care for the health of all people in Samoa. The Samoa Health Sector Plan 2008–2018 prioritized NCD

prevention through health promotion programs and focused on addressing smoking, alcohol, nutrition, and physical activity as key behaviors to modify in the population [19]. This builds off decades of collaboration with the Ministry of Education, Sports, and Culture, and the Ministry of Women, Social Development, and Communities, to champion health promoting school guidelines and health education initiatives for children and adolescents [19]. Starting in 2012, School Nutrition Standards were developed to guide and monitor the compliance of schools to support healthy eating [21]. A school fruit tree program was implemented to encourage the planting and eating of local fruits and vegetables, while the *Aiga ma Nu'u Manuia* program promoted gardening in village communities [19]. In 2015, the Samoa Ministry of Health adopted and piloted the World Health Organization's Package of Essential NCD (PEN) interventions, known as PEN *Fa'a Samoa*, to include community participation, to improve outreach services, and to increase village-based monitoring of NCD risk factors, especially in children and women [22].

Thus far, there have been two waves of the GSHS conducted among children attending schools across Samoa in 2011 and 2017 [23, 24]. The use of the existing GSHS data is an opportunity to better understand differences in NCD risk and protective factors at a population-level in this local context and, in turn, inform actions and decision-making for NCD-focused strategies in the country. Our objective was, therefore, to use publicly available GSHS data from Samoa to provide cross-sectional, and trend estimates for the prevalence of NCD risk and protective factors among school-attending children and adolescents. Risk factors included overweight/obesity, drinking carbonated soft drinks, hunger, sedentary activity, absenteeism, substance use, poor hand hygiene, poor emotional and mental health including suicide ideation. Protective factors included eating fruits and vegetables daily, physical activity, brushing teeth, and community support, such as other students being kind and helpful, or having a parent or guardian who understands their troubles.

## Materials and methods

### Study design and data source

This retrospective study used deidentified data from the GSHS in 2011 and 2017. GSHS methods have been previously described by the WHO and CDC [10, 25]. The study used a two-stage cluster sampling design to recruit a nationally representative sample of children in school years 8–13 [10]. In Samoa, schools were first sampled using a probability proportionate to the school enrollment of children, and then, classes from the school year that had the highest proportion of 13-15-year-olds were randomly sampled from participating schools [10]. All children in those sampled classes were eligible for the survey and invited to complete an anonymous self-report computer-scannable survey form during school hours [9, 10].

In Samoa, 83% of the sampled schools participated in 2011 and 94% in 2017 [23, 24]. In the second stage, the response rate among children invited to complete the survey was 96% in 2011 and 63% in 2017. After accounting for non-participating schools, the overall response rate was 79% 2011 and 59% in 2017 [23, 24]. The final analytic sample included 2,418 children in 2011 and 1,955 children in 2017.

All de-identified datasets, questionnaires, codebooks, and other study documents were made available to the public approximately two years after data collection, and they were downloaded from the WHO NCD microdata repository and CDC website for this analysis [9, 10, 25–27]. The study received a Category 4 research exemption from the Yale Human Research Protection Program (IRB # 2000032709), given that the data used were unidentifiable and accessible within the public domain, and was approved by the Health Research Committee at the Samoa Ministry of Health.

## Measures

The variables used for this analysis and questions asked in the GSHS are described in S1 Table. The GSHS questionnaire was found to have good internal validity in a previous validation study in another Pacific Island country [28] and other settings [29].

Categories for the study variables were determined *a priori*, based on CDC analysis recommendations [25] and those previously used in GSHS reports and other studies, to allow for better comparability within and between countries [30–33]. Body weight and height were self-reported and used to calculate body mass index (BMI). Children with a BMI Z-score > +1 standard deviation based on WHO 2007 child references for age and sex were classified as having overweight/obesity [10, 34]. Adequate fruit and vegetable consumption was defined as two or more and three or more servings per day, respectively. Any consumption of sugar-sweetened beverages was based on drinking carbonated drinks, like Coca-Cola, at least 1 or more times per day during the past month. Children who reported going hungry 'mostly' or 'always' in the past month were considered to have 'gone hungry'. Children who were physically active for at least 60 minutes per day for 5+ days per week were considered to engage in "physical activity". Sedentary behavior was defined as spending 3 or more hours per day sitting down. For physical education (PE), children who went to PE class at least 5 days per week were considered to do adequate PE. The current smoking status of cigarettes or other tobacco products in the past month was based on child-report of smoking cigarettes or using any other form of tobacco such as e-cigarettes or chewed smokes for at least one day. For alcohol drinking, children reported the number of drinks in the past month and whether they drank so much alcohol that they were really drunk during their life. Based on the reported frequency of oral and hand hygiene practices, children were categorized into brushing their teeth at least 1 time per day, or not always washing their hands with soap, before eating, or after toilet use. For emotional and mental health, children reported ever being bullied, in a physical fight, feeling lonely, worried that they could not sleep at night, having no close friends, and suicidal ideation in the past month. Children who reported at least 1 day that they missed class or school without permission in the past month were compared to those who never missed. In terms of community protective factors, children reported mostly or always having parents/guardians check homework, others being kind and helpful, parents/guardians understanding troubles, and parents/guardians knowing what they do.

## Statistical analyses

We proceeded with statistical analyses that accounted for the two-stage cluster sampling methods used for the recruitment of schools and classrooms, following recommendations and guidance from the CDC for the analysis of GSHS data [10, 25]. Weight, stratum, and primary sampling unit (PSU) variables were provided with each dataset in 2011 and 2017 to account for the cluster sampling method, missing data, and to adjust for minor within-country demographic differences between participants and the total population of school-attending children in the country [25].

The sample sizes were presented to reflect the number of children who completed each question in the GSHS. Weighted percentages were reported to describe sample characteristics (e.g., age group, sex, and school year) and cross-sectional prevalences of various health risk behaviors and protective factors. GSHS data were weighted to adjust for school and nonresponse and to make the data representative of the population of children who attended school from which the sample was drawn [25]. To assess differences in health risk and protective factors between 2011 and 2017, we followed CDC recommendations to analyze GSHS data by adapting statistical code that was previously developed for the Youth Risk Behavior

Surveillance System [25, 35]. Multivariable logistic regressions were performed for females and males separately with each factor regressed on the year of study (2011 or 2017), age group, and school year to minimize confounding and selection bias. Interactions of study year with age group were not added to the models because no differences were observed (p > 0.05).

Two-sided 95% confidence intervals (CIs) and p-values were reported, after adjusting for the multistage, stratified cluster sample design of the GSHS. Following CDC recommendations for trend analyses and previous research using multiple waves of GSHS data [14, 15, 35], we considered 95% confidence intervals excluding an odds ratio of 1 and p-values < 0.05 to represent differences in risk or protective factors between 2011 and 2017; however, the OR estimate cannot be directly interpreted as the magnitude of difference between 2011 and 2017. All analyses were conducted in SAS version 9.4. (SAS Institute, North Carolina, USA).

## Results

### Sample characteristics

Two waves of the Samoan GSHS sample included 4,373 children (51.60% in 2011 and 52.19% in 2017 were female), with a median age of 15 years and the majority of whom were enrolled in school years 9–10 (Table 1).

**Overweight and obesity.** After adjusting for age group and school year, the prevalence of overweight/obesity was consistently high in 2011 and 2017 among females (59.12% vs. 64.29%, p = 0.428, Table 2). Overweight/obesity was prevalent among 44.21% of males in 2011 and this was higher in 2017 (53.65%, p = 0.039, Table 3).

**Table 1. Sample characteristics of participating students during 2011 and 2017 Global School-based Student Health Surveys in Samoa (N = 4,373)\*.**

| Characteristics | 2011 (n = 2,418) | 2017 (n = 1,955) |
|---|---|---|
| | n (%) | n (%) |
| Age group (years) | | |
| ≤11 | 50 (2.45) | 47 (3.22) |
| 12 | 84 (3.22) | 98 (7.20) |
| 13 | 513 (19.06) | 237 (13.72) |
| 14 | 981 (39.85) | 368 (17.88) |
| 15 | 622 (27.98) | 355 (17.15) |
| ≥16 | 155 (7.43) | 819 (40.82) |
| Sex | | |
| Female | 1376 (51.60) | 1197 (52.19) |
| Male | 970 (48.40) | 707 (47.81) |
| School level | | |
| Year 8 | — | 271 (20.73) |
| Years 9 | 1400 (51.77) | 447 (20.13) |
| Year 10 | 939 (45.52) | 378 (18.52) |
| Year 11 | 13 (0.91) | 278 (15.49) |
| Year 12 | 26 (1.80) | 276 (14.16) |
| Year 13 | — | 274 (10.96) |

\* N reported reflects the sample that completed the Global School-based Student Health Survey and will not sum to the total due to missing data. Percentages are weighted and adjusted for the multistage stratified cluster sample design of the survey and student nonresponse to make the data representative of the population of students from which the sample was drawn.

**Table 2. Health risk and protective factors in 2011 and 2017 among females in Samoa.**

| | 2011 | 2017 | Difference | P |
|---|---|---|---|---|
| | n (%)* | n (%)* | Adjusted OR** (95%CI) | |
| **Overweight/obesity** (>+1SD from the median for BMI by age and sex) | 561 (59.12) | 725 (64.29) | 1.08 (0.89–1.31) | 0.428 |
| **Dietary behavior** | | | | |
| Ate Fruits (2+ times/ day) | 646 (47.50) | 605 (51.57) | 1.23 (0.97–1.57) | 0.085 |
| Ate Vegetables (3+ times/ day) | 505 (38.15) | 383 (32.87) | 0.78 (0.64–0.96) | 0.019 |
| Drank carbonated soft drinks (1+ times/day) | 721 (53.25) | 775 (64.93) | 1.56 (1.27–1.92) | <0.001 |
| Went hungry (mostly/always) | 483 (36.01) | 144 (12.31) | 0.26 (0.18–0.38) | <0.001 |
| **Physical Activity and Sedentary Behaviors** | | | | |
| Active 60+ mins/day for 5+ of past 7 days | 265 (21.58) | 348 (29.94) | 1.40 (1.04–1.89) | 0.028 |
| Time sitting 3+ hours/day | 412 (32.89) | 300 (25.26) | 0.59 (0.42–0.84) | 0.004 |
| Attended physical education classes 5+days/week during the school year | 199 (15.98) | 159 (14.32) | 0.95 (0.62–1.47) | 0.823 |
| **Substance Use** | | | | |
| Smoked cigarettes in the past month | 340 (26.99) | 87 (7.37) | 0.19 (0.12–0.31) | <0.001 |
| Smoked other tobacco products in the past month | 439 (32.20) | 52 (4.78) | 0.12 (0.07–0.20) | <0.001 |
| Drank any alcohol in the past month | 334 (27.34) | 105 (9.08) | 0.24 (0.16–0.36) | < .0001 |
| Drunk in life time | 310 (26.49) | 67 (6.20) | 0.18 (0.12–0.27) | <0.001 |
| **Oral and Hand Hygiene** | | | | |
| Brushed teeth (1 times/day) | 1164 (85.80) | 1148 (96.33) | 3.64 (1.99–6.67) | <0.001 |
| Wash hands with soap (not always) | 225 (16.86) | 127 (10.70) | 0.66 (0.50–0.88) | 0.006 |
| Wash hands before eating (not always) | 161 (11.85) | 184 (15.11) | 1.30 (0.96–1.78) | 0.094 |
| Wash hands after toilet use (not always) | 206 (15.26) | 65 (5.76) | 0.42 (0.29–0.62) | <0.001 |
| **Emotional and Mental Health** | | | | |
| Missed class or school without permission (absenteeism) | 590 (49.41) | 369 (33.40) | 0.45 (0.32–0.61) | <0.001 |
| Bullied | 820 (69.55) | 365 (33.27) | 0.23 (0.16–0.35) | <0.001 |
| In physical fight | 833 (63.33) | 440 (38.53) | 0.38 (0.29–0.50) | <0.001 |
| Felt lonely (mostly/always) | 303 (22.49) | 109 (9.39) | 0.36 (0.27–0.48) | <0.001 |
| Worried that could not sleep at night (mostly/always) | 384 (28.55) | 111 (8.94) | 0.25 (0.18–0.34) | <0.001 |
| Had no close friends | 732 (55.65) | 246 (21.09) | 0.22 (0.15–0.33) | <0.001 |
| Seriously considered attempting suicide in the past 12 months | 342 (29.51) | 256 (22.09) | 0.66 (0.51–0.86) | <0.001 |
| Made plan to attempt suicide in the past 12 months | 432 (35.06) | 272 (23.50) | 0.58 (0.45–0.76) | <0.001 |
| Attempted suicide in the past 12 months | 193 (15.75) | 95 (7.94) | 0.52 (0.35–0.78) | 0.002 |
| **Community protective factors (reported as most/always)** | | | | |
| Other students were kind and helpful | 441 (36.42) | 429 (37.36) | 0.93 (0.71–1.22) | 0.601 |
| Parents/guardians check homework | 579 (47.52) | 598 (52.21) | 1.07 (0.82–1.41) | 0.613 |
| Parents/guardians understand troubles | 423 (35.78) | 326 (28.51) | 0.71 (0.55–0.90) | 0.005 |
| Parents/guardians know what you do | 426 (35.20) | 363 (32.06) | 0.88 (0.67–1.16) | 0.363 |

* N reported reflects the sample that completed the Global School-based Survey. Percentages are weighted and adjusted for the multistage stratified cluster sample design of the survey.

** Odds ratios (OR) and corresponding 95% confidence intervals were estimated from logistic regression models adjusted for the child age group, school year, and the multistage stratified cluster sample design of the survey. Per CDC analysis guidance (35), the OR estimate cannot be directly interpreted as the magnitude of difference between 2011 and 2017. Any OR >1 or OR <1, with a 95% CI not including 1, and p<0.05 was interpreted as a meaningful difference.

**Dietary behaviors.** Half of the children ate fruits at least twice per day (47.50% in females and 52.43% in males) in 2011, and this was similar in 2017 (all p <0.05). A little more than a third of the children ate vegetables at least three times per day (38.15% in females and 37.72% in males) in 2011; the proportion of children reaching this threshold was lower among females

**Table 3. Health risk and protective factors in 2011 and 2017 among males in Samoa.**

| | 2011 | 2017 | Change | P |
|---|---|---|---|---|
| | n (%)* | n (%)* | Adjusted OR (95%CI) | |
| **Overweight/obesity** (>+1SD from the median for BMI by age and sex) | 296 (44.21) | 361 (53.65) | 1.41 (1.02–1.94) | 0.039 |
| **Dietary behavior** | | | | |
| Ate Fruits (2+ times/ day) | 494 (52.43) | 358 (50.23) | 0.87 (0.68–1.13) | 0.292 |
| Ate Vegetables (3+ times/ day) | 357 (37.72) | 246 (35.64) | 0.92 (0.72–1.17) | 0.469 |
| Drank carbonated soft drinks (1+ times/day) | 510 (54.61) | 433 (61.08) | 1.24 (0.98–1.57) | 0.076 |
| Went hungry (mostly/always) | 342 (36.26) | 79 (12.59) | 0.29 (0.19–0.46) | <0.001 |
| **Physical Activity and Sedentary Behaviors** | | | | |
| Active 60+ mins/day for 5+ of past 7 days | 175 (20.06) | 228 (33.34) | 1.69 (1.26–2.27) | <0.001 |
| Time sitting 3+ hours/day | 378 (44.58) | 191 (30.13) | 0.48 (0.33–0.68) | <0.001 |
| Attended physical education classes 5+days/week during the school year | 102 (11.74) | 115 (18.06) | 1.70 (0.91–3.16) | 0.092 |
| **Substance Use** | | | | |
| Currently smoked cigarettes in the past month | 370 (44.13) | 91 (13.40) | 0.18 (0.13–0.25) | <0.001 |
| Currently smoked other tobacco products in the past month | 45 (48.23) | 77 (11.88) | 0.17 (0.12–0.23) | <0.001 |
| Drank any alcohol in the past month | 380 (45.17) | 109 (15.60) | 0.21 (0.15–0.28) | <0.001 |
| Drunk in life time | 364 (48.01) | 71 (10.90) | 0.13 (0.09–0.18) | <0.001 |
| **Oral and Hand Hygiene** | | | | |
| Brushed teeth (≥1 time/day) | 727 (76.51) | 665 (94.99) | 5.78 (3.77–8.85) | <0.001 |
| Wash hands with soap (not always) | 196 (21.24) | 125 (18.82) | 0.86 (0.64–1.15) | 0.296 |
| Wash hands before eating (not always) | 161 (17.26) | 160 (23.11) | 1.26 (0.93–1.70) | 0.132 |
| Wash hands after toilet use (not always) | 185 (19.66) | 58 (8.15) | 0.38 (0.27–0.55) | <0.001 |
| **Emotional and Mental Health** | | | | |
| Missed class or school without permission (absenteeism) | 499 (62.23) | 250 (40.28) | 0.43 (0.30–0.61) | <0.001 |
| Bullied | 640 (79.16) | 261 (41.29) | 0.22 (0.17–0.28) | <0.001 |
| In physical fight | 683 (73.72) | 387 (57.28) | 0.51 (0.40–0.65) | <0.001 |
| Felt lonely (mostly/always) | 220 (23.80) | 63 (8.58) | 0.30 (0.21–0.43) | <0.001 |
| Worried that could not sleep at night (mostly/always) | 247 (27.03) | 71 (9.51) | 0.27 (0.19–0.39) | <0.001 |
| Had no close friends | 620 (68.21) | 151 (23.35) | 0.14 (0.11–0.19) | <0.001 |
| Seriously considered attempting suicide in the past 12 months | 276 (37.75) | 148 (23.29) | 0.49 (0.37–0.66) | <0.001 |
| Made plan to attempt suicide in the past 12 months | 352 (45.30) | 139 (20.98) | 0.33 (0.25–0.43) | <0.001 |
| Attempted suicide in the past 12 months | 143 (17.36) | 67 (11.36) | 0.71 (0.50–1.02) | 0.066 |
| **Community protective factors (reported as most/always)** | | | | |
| Other students were kind and helpful | 254 (30.78) | 230 (34.33) | 1.12 (0.86–1.46) | 0.399 |
| Parents/guardians check homework | 289 (35.85) | 277 (42.28) | 1.29 (0.995–1.67) | 0.054 |
| Parents/guardians understand troubles | 238 (30.05) | 143 (21.03) | 0.62 (0.46–0.83) | 0.001 |
| Parents/guardians know what you do | 209 (25.46) | 156 (24.61) | 1.01 (0.75–1.35) | 0.969 |

* N reported reflects the sample that completed the Global School-based Survey. Percentages are weighted and adjusted for the multistage stratified cluster sample design of the survey.

** Odds ratios (OR) and corresponding 95% confidence intervals were estimated from logistic regression models adjusted for the child age group, school year, and the multistage stratified cluster sample design of the survey. Per CDC analysis guidance (35), the OR estimate cannot be directly interpreted as the magnitude of difference between 2011 and 2017. Any OR >1 or OR <1.0, with a 95% CI not including 1, and p<0.05 was interpreted as a meaningful difference.

in 2017 (p = 0.019). The majority of females (53.25%) and males (54.61%) drank carbonated soft drinks at least once a day in 2011 and this was higher among females in 2017 (64.93%, p<0.001), but not males (p = 0.076). One in three females (36.01%) reported being 'mostly or always' hungry in 2011 and this was lower in 2017 (12.31%, p<0.001), and this trend was similar in males (36.26% in 2011 and 12.59% in 2017, p<0.001).

**Physical activity and sedentary activities.** Compared to 2011, more children reported at least 5 days of physical activity in the past week for at least 60 minutes a day in 2017 among females (21.58% vs. 29.94%, p = 0.028) and males (20.06% vs. 33.34%, p<0.001). Sitting for three or more hours per day was higher among females in 2011 (32.89%) than in 2017 (25.26%, p = 0.004) and this was similar among males (44.58% vs 30.13%, p<0.001). The prevalence of females and males attending PE classes was similar between 2011 and 2017 (both p>0.05).

**Substance use.** Males had the highest prevalence of cigarette smoking (44.13%) and alcohol use (45.17%) in 2011, compared to (13.40% and 15.60%, respectively both p <0.05). Few females and males reported smoking other tobacco products (4.78% and 11.88%, respectively) and being drunk in their lifetime (6.20% and 10.90%, both p <0.001).

**Oral and hand hygiene.** Tooth brushing at least once per day was highly prevalent in 2011 (85.80% in females and 76.51% in males) and this was higher in 2017 (96.33% in females and 94.99% in males, all p<0.001). Not always hand washing with soap declined among females from 16.86% in 2011 to 10.70% (p = 0.006), but the prevalence remained the same for males (21.24% vs. 18.82%, p = 0.296). Some students reported not always washing their hands before eating and this was similar between 2011 and 2017 among females (11.85% vs. 15.11% p = 0.094) and males (17.26% vs. 23.11%, p = 0.132). Few females reported not always washing hands after toilet use, with prevalence decreasing from 15.26% to 5.76% (p<0.001) and this was similar in males (19.66% vs. 8.15%, p<0.001).

**Emotional and mental health.** Many students reported being bullied (74.75%), or in a physical fight (68.74%) in 2011, which were markedly lower in 2017 in the total sample (S2 Table). Bullying was prevalent in 2011 (69.55%) and lowered in 2017 among females (33.27%, p<0.001); this was also observed in males (79.16% in 2011 to 59.30% in 2017, p<0.001).

Some students reported feeling lonely and worrying about something such that they could not fall asleep in 2011 and this was lower in 2017. Although half of females (55.65%) and most males (68.21%) reported having no close friends in 2011, the prevalence was lower in 2017 (21.09% and 23.35%, both p<0.001), respectively. Suicidal ideation, seriously thinking about making a suicide plan, was reduced among females and males (all p<0.001), but the prevalence of attempting suicide was lower only among females (15.75% in 2011 vs. 7.94%, p = 0.002).

**Community protective factors.** Peer support was reported by some students and remained at similar levels between 2011 and 2017; in 2017, 37.36% of females and 34.33% of males reported that other students were kind and helpful. While parental or guardian supervision to check homework and really know what students were doing with their free time did not appear to differ, fewer reported that parents or guardians mostly or always understood their troubles in 2017 compared to 2011 among females (35.78% to 28.51%, p = 0.005) and males (30.05% to 21.03%, p = 0.001).

## Discussion

The high and rising prevalence of overweight/obesity among Samoan school-aged children calls for continued obesity-related NCD prevention and treatment efforts. While more females were affected by overweight/obesity, a greater rise in prevalence was observed in males. Declines in reported sitting for long periods of time per day, smoking cigarettes, using other tobacco products, and drinking alcohol suggest some progress towards meeting the NCD Global Action Plan goals to reduce sedentary activity and harmful substance use by 2030. The high prevalence of bullying, physical fighting especially among males, having no close friends, and reports of suicide ideation are concerning and highlight the need to better prioritize these risk factors in global action plans. Intensified policies and interventions focused on addressing

emotional and mental wellness–which may include culturally-rooted and community-based programs–are critical to reduce NCD risk and advance health equity for children in Samoa.

While the school environment can support NCD prevention and interventions, its influence on modifiable risk factors and subsequent health outcomes among children is not clear in Samoa. Schools may help to promote healthy eating behaviors and provide safe spaces for physical activity to prevent the development of risk behaviors that may predispose them to greater NCD risk [36, 37]. Although the differences in dietary behaviors did not reach statistical significance among males between two waves of GSHS, the consumption of vegetables lowered and consumption of carbonated soft drinks increased across the sample in 2017 compared to 2011. These differences may be partly explained by the increasing costs of produce on island, and greater availability of low-cost imported food and drinks [38]. Continued nutrition education programs in primary schools across Samoa, such as the 'Eat the Rainbow' campaign [39], were also anticipated to encourage healthier eating behaviors. For physical activity, more children reported being active for 60 minutes or more in 2017 compared to 2011. These activities were likely occurring outside of schools because the level of attendance in physical education classes was similar between 2011 and 2017. This is possible considering that physical activities are often based in communities and positive connections to their community and experiences with others may enable children to develop a balanced set of physical, social, emotional and cognitive skills to thrive beyond school [40]. The protective community factors may change over time and how it is related to other NCD risk behaviors requires further investigation.

There were notable differences in harmful substance use and improved hand hygiene between 2011 and 2017, which coincided with several public health campaigns, policies, and programs implemented by divisions at the Ministry of Health, including the National Health Programs and Health Enforcement and Protection [41]. For the past two decades, a multi-sector National Tobacco Control Committee has advocated for policies and implemented programs related to the monitoring and enforcement of tobacco control in Samoa [19]. With the implementation of the School Tobacco Control Program in 2014, there was greater accessibility to smoke-free spaces and cessation services for students [42]. Around the time of the administration of the first GSHS in Samoa, the Liquor Act of 2011 established a national legal minimum age of 21 years for the sale of alcoholic beverages [43] and this was expected to reduce alcohol consumption among school-aged children and adolescents. Although no national oral health policy exists [44], national health promotion activities were implemented to encourage brushing teeth and hand washing nearly every year [45]. To prevent and further reduce NCD risk factors, public health services must continue to provide children with opportunities to develop positive behaviors for healthier lifestyles.

The alarmingly high prevalence of perceived bullying, physical fighting especially among males, loneliness, and reports of suicide ideation in Samoan children calls for more action focused on the promotion of mental health literacy and wellness as part of NCD prevention efforts. In a 1995 Apia Urban Youth Survey 49% of youth believed that suicide was the most serious problem they faced [46] and the data presented here argue that this may still be the case. Our study findings are similar to the 2013 Youth Risk Behavior Survey in American Samoa, where 38% of students in years 9–12 reported feeling sad or hopeless almost every day during the past two weeks, and 23% had made a suicide plan in the last year [47]. A more recent (2021) survey conducted in the territory by the Empowering Pacific Islander Communities non-governmental organization, which screened 1125 high school students across three high schools identified that one in three adolescents seriously thought about suicide and nearly 30% endorsed having attempted to die by suicide in the past year [48]. In Samoa, participants from the Youth and Mental Health project shared their values of Samoan culture, as a part of

their sense of self and dignity, and recognized how their selves are connected to the support from within families and beyond in their villages and/or church organizations [46]. Adolescents in American Samoa perceived structural, social, interpersonal, and self-stigma of mental illness [49], which may impede care-seeking behaviors in their community and require further investigation. Continued development of interventions and research approaches rooted in culturally specific constructs and context are critical, particularly in the Pacific communities where family and religion influence the lives of children [50].

## Limitations and strengths

Several limitations of the GSHS protocol and data require cautious interpretation of the results. Given that the sampling approach focused on a study population that was representative of all children attending school at the time of the survey administration, the cross-sectional and trend estimates may not be generalizable to all children and adolescents in Samoa (who were not attending school, but in the same age range). There is the possibility that the prevalence estimates are conservative for some risk and protective factors. If there exists reporting bias by sex (females may underestimate weight more than males) and weight status (those who are larger in body size may underestimate weight more than those who are not) [51], overweight/obesity prevalence was likely underestimated. On the other hand, self-reported weight may not be a true reflection of their weight because scales are not commonly owned, and weight is not well monitored in this setting. As with any self-reported data, there is the potential for recall bias and social desirability bias to be introduced because students are asked to recall health behaviors from different periods of time (for example, during the past 7 days versus during the past 12 months) and may be inclined to provide responses that are acceptable by social norms and culture in Samoa. Considering the stigma around mental illness, structural barriers to accessing mental health services, and poor mental health literacy in Pacific communities [46, 49, 52], the estimates may be conservative, and future work should examine emotional and mental health where data are available, like GSHS. Following 2011, the introduction of health promotion programs and policies to support cessation efforts of tobacco and alcohol use may also act to increase the social desirability in responding to health-related behaviors, and in turn, this would result in a lower prevalence reported in 2017. While trend estimates are presented as differences between 2011 and 2017, we were not able to infer any causal relationships. Although there were response rate differences in the age groups and school years sampled in 2011 and 2017, the analysis was weighted to account for the cluster sampling method and adjust for minor within-country demographic differences between participants, following CDC analysis guidance [25]. Importantly, these findings align with the guiding principles of the National Health Promotion Policy and vision that "all individuals, families, and communities in Samoa are empowered to promote their health and well-being to live healthier, happy, and productive" [45].

## Conclusion

The high and increasing prevalence of NCD risk factors highlight the importance of directing national public health programming and interventions towards positive behavior changes, as well as, prioritizing emotional and mental wellness in Samoa. Future efforts to continue the GSHS will help to document temporal trends of NCD risk factors and protective factors and may be used as a tool for monitoring the effectiveness of public health campaigns and preventative programs.

## Supporting information

**S1 Table. Summary of variables and questions from the Global student-based health survey (GSHS) in Samoa.**
(DOCX)

**S2 Table. Health risk behaviors among all students in 2011 and 2017.**
(DOCX)

**S1 Checklist. STROBE checklist.**
(DOCX)

## Acknowledgments

This paper uses data from the Global School-Based Student Health Survey (GSHS), which is supported by the World Health Organization (WHO) and the United States Centers for Disease Control and Prevention (CDC). We are grateful to the students, families, and schools who participated in the GSHS, Mr. Deuel Meredith who was the coordinator in Samoa in 2011, at the Ministry of Health (especially Faaifoaso Moala, Lautala Malaga and Taugofie Aleki, and Analosa Manuele-Magele who were part of the survey implementation team) and Ministry of Education School and Community in Samoa for making the GSHS possible, and to WHO and CDC for making the public data available for analysis.

## Author Contributions

**Conceptualization:** Courtney C. Choy, Siufaga Simi, Take Naseri, Nicola L. Hawley.

**Data curation:** Siufaga Simi.

**Formal analysis:** Courtney C. Choy, Yasmmyn D. Salinas.

**Funding acquisition:** Courtney C. Choy, Take Naseri, Nicola L. Hawley.

**Investigation:** Courtney C. Choy, Siufaga Simi, Christina Soti-Ulberg, Take Naseri, Yasmmyn D. Salinas, Nicola L. Hawley.

**Methodology:** Courtney C. Choy, Siufaga Simi, Yasmmyn D. Salinas.

**Project administration:** Siufaga Simi, Take Naseri.

**Resources:** Courtney C. Choy, Siufaga Simi, Christina Soti-Ulberg, Take Naseri, Nicola L. Hawley.

**Software:** Courtney C. Choy.

**Supervision:** Take Naseri, Yasmmyn D. Salinas, Nicola L. Hawley.

**Visualization:** Courtney C. Choy.

**Writing – original draft:** Courtney C. Choy, Siufaga Simi, Yasmmyn D. Salinas, Nicola L. Hawley.

**Writing – review & editing:** Courtney C. Choy, Siufaga Simi, Christina Soti-Ulberg, Take Naseri, Yasmmyn D. Salinas, Nicola L. Hawley.

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
