## [Decision Letter · Decision Letter 0]

9 May 2024

PGPH-D-24-00472

Noncommunicable disease risk behaviors and protective factors among children in Samoa between 2011 and 2017: Retrospective trend analysis of global school-based health surveys

Dear Dr. Choy,

Thank you for submitting your manuscript to PLOS Global Public Health. After careful consideration, we feel that it has merit but does not fully meet PLOS Global Public Health’s publication criteria as it currently stands. Therefore, we invite you to submit a revised version of the manuscript that addresses the points raised during the review process.

EDITOR: Dear Author, please clarify and make the changes based on the reviewer's comments in the PDF document.

The decision of this manuscript is justified based on PLOS Global Public Health’s publication criteria and not on its novelty or perceived impact.

We look forward to receiving your revised manuscript.

Kind regards,

Zulkarnain Jaafar

Academic Editor

Journal Requirements:

2. We ask that a manuscript source file is provided at Revision. Please upload your manuscript file as a .doc, .docx, .rtf or .tex.

Additional Editor Comments (if provided):

Reviewers' comments:

Reviewer's Responses to Questions

**Comments to the Author**

1. Does this manuscript meet PLOS Global Public Health’s publication criteria? Is the manuscript technically sound, and do the data support the conclusions? The manuscript must describe methodologically and ethically rigorous research with conclusions that are appropriately drawn based on the data presented.

Reviewer #1: Yes

Reviewer #2: Yes

2. Has the statistical analysis been performed appropriately and rigorously?

Reviewer #1: Yes

Reviewer #2: Yes

3. Have the authors made all data underlying the findings in their manuscript fully available (please refer to the Data Availability Statement at the start of the manuscript PDF file)?

Reviewer #1: Yes

Reviewer #2: Yes

4. Is the manuscript presented in an intelligible fashion and written in standard English?

Reviewer #1: Yes

Reviewer #2: Yes

5. Review Comments to the Author

Reviewer #1: This is an interesting paper. The authors have successfully established the need for research, and have focused on a pertinent public health problem. Data analysis is robust and findings have been well interpreted.

Reviewer #2: attached

6. PLOS authors have the option to publish the peer review history of their article (what does this mean?). If published, this will include your full peer review and any attached files.

**Do you want your identity to be public for this peer review?** For information about this choice, including consent withdrawal, please see our Privacy Policy.

Reviewer #1: No

Reviewer #2: No

---

## [Editor Report · Decision Letter 1]

20 May 2024

Noncommunicable disease risk behaviors and protective factors among children in Samoa: Retrospective trend analysis of global school-based health surveys in 2011 and 2017

PGPH-D-24-00472R1

DearDr Choy,

We are pleased to inform you that your manuscript 'Noncommunicable disease risk behaviors and protective factors among children in Samoa: Retrospective trend analysis of global school-based health surveys in 2011 and 2017' has been provisionally accepted for publication in PLOS Global Public Health.

Best regards,

Zulkarnain Jaafar

Academic Editor